# Hypericin Inhibit Alpha-Coronavirus Replication by Targeting 3CL Protease

**DOI:** 10.3390/v13091825

**Published:** 2021-09-14

**Authors:** Yue Zhang, Huijie Chen, Mengmeng Zou, Rick Oerlemans, Changhao Shao, Yudong Ren, Ruili Zhang, Xiaodan Huang, Guangxing Li, Yingying Cong

**Affiliations:** 1Veterinary Pathology Laboratory, College of Veterinary Medicine, Heilongjiang Key Laboratory for Animal and Comparative Medicine, Northeast Agricultural University, Harbin 150030, China; zy_virus@sina.com (Y.Z.); chenhuijie1983@163.com (H.C.); 18738672357@163.com (M.Z.); changhaos1994@163.com (C.S.); pathology4321@163.com (R.Z.); hxd790125@neau.edu.cn (X.H.); 2College of Pharmaceutical Engineering, Jilin Agriculture Science and Technology University, Jilin 132101, China; 3Department of Drug Design, University of Groningen, 9713 AV Groningen, The Netherlands; r.oerlemans@rug.nl; 4Department of Computer Science and Technology, College of Electrical and Information Technology, Northeast Agricultural University, Harbin 150030, China; ydren@neau.edu.cn; 5Department of Biomedical Sciences of Cells and Systems, University of Groningen, University Medical Center Groningen, 9713 AV Groningen, The Netherlands

**Keywords:** α-CoV, PEDV, hypericin, 3CL protease, TGEV

## Abstract

The porcine epidemic diarrhea virus (PEDV) is an *Alphacoronavirus* (α-CoV) that causes high mortality in infected piglets, resulting in serious economic losses in the farming industry. Hypericin is a dianthrone compound that has been shown as an antiviral activity on several viruses. Here, we first evaluated the antiviral effect of hypericin in PEDV and found the viral replication and egression were significantly reduced with hypericin post-treatment. As hypericin has been shown in SARS-CoV-2 that it is bound to viral 3CLpro, we thus established a molecular docking between hypericin and PEDV 3CLpro using different software and found hypericin bound to 3CLpro through two pockets. These binding pockets were further verified by another docking between hypericin and PEDV 3CLpro pocket mutants, and the fluorescence resonance energy transfer (FRET) assay confirmed that hypericin inhibits the PEDV 3CLpro activity. Moreover, the alignments of α-CoV 3CLpro sequences or crystal structure revealed that the pockets mediating hypericin and PEDV 3CLpro binding were highly conserved, especially in transmissible gastroenteritis virus (TGEV). We then validated the anti-TGEV effect of hypericin through viral replication and egression. Overall, our results push forward that hypericin was for the first time shown to have an inhibitory effect on PEDV and TGEV by targeting 3CLpro, and it deserves further attention as not only a pan-anti-α-CoV compound but potentially also as a compound of other coronaviral infections.

## 1. Introduction

Coronaviruses (CoVs) are viruses that cause numerous pathologies in humans and other mammals, including respiratory, enteric, hepatic, and neurological diseases, with varying severities [1]. CoVs are divided into four genera, i.e., *Alphacoronavirus* (α-CoV), *Betacoronavirus* (β-CoV), *Gammacoronavirus* (γ-CoV), and *Deltacoronavirus* (δ-CoV). Porcine epidemic diarrhea virus (PEDV) is an α-CoV and the causative pathogen of porcine epidemic diarrhea (PED), which is mainly characterized by vomiting, watery diarrhea, severe dehydration, and weight loss [2]. PEDV infections have a substantially detrimental effect on the swine industry with high morbidity and mortality rates, especially in suckling piglets. In the past 10 years, new outbreaks of PED have swept through many countries in Asia and America, leading to huge losses in the global pig industry [3,4]. Moreover, the transmission of PED is mainly through a fecal–oral route from the feces of virus-carrying pigs to others and shows seasonality, has more outbreaks in winter and spring, and is scattered in summer and autumn [5]. Hence, it becomes important to develop a drug/vaccine to fight against PED.

CoVs are enveloped positive single-stranded RNA viruses; the genome is approximately 30 kb and encodes four structural proteins, accessory proteins, and two overlapping open reading frames (ORFs), ORF1a and ORF1b, which are then translated into two large polyproteins, pp1a and pp1ab. These two polyproteins are processed into 15 or 16 nonstructural proteins (nsps) by multiple viral proteinases present in their sequences that are essential for CoV replication [6]. α-CoVs have three viral proteases that are involved in the self-cleavage of pp1a and pp1ab: two papain-like proteases (PLpro), present with nsp3, a 3C-like proteinase (3CLpro) or Mpro (hereafter 3CLpro), localized in nsp5. 3CLpro digests the viral polyproteins at 11 sites and further processes them into 12 nsps. Moreover, the 3CLpro is highly conserved among CoVs with an average similarity of 51.0% [7], and it appears like a high-profile target for antiviral drug discovery in severe acute respiratory syndrome coronavirus 2 (SARS-CoV-2) [8,9,10].

The usage of veterinary antiviral drugs in food-producing animals has the potential to generate residues in animal-derived products (meat, milk, eggs, and honey) and poses a health hazard to the consumer. In 2019, the Ministry of Agriculture and Rural Affairs of the People’s Republic of China banned the usage of some antiviral drugs in food-producing animals [11]. Thus, it is urgent to explore other ways to fight against animal viruses; one of the possibilities is to take advantage of traditional antiviral herbs that show a lower risk of drug resistance. Interestingly, more and more reports have shown that herbs could effectively inhibit the replication of various viruses [12]. Hypericin is one of the most biologically active dianthrone compounds extracted from St John’s Wort (*Hypericum perforatum* L.) [13] that is widely used in antiviral [14,15,16], antitumor [17,18] and antidepressant fields [19,20]. In particular, hypericin has shown strong antiviral activities against uncoated RNA viruses, e.g., human immunodeficiency virus (HIV) and hepatitis C virus (HCV) [14,21,22]. Moreover, our recent study showed an inhibition effect of hypericin in a γ-CoV, infectious bronchitis virus (IBV), through the inhibition of apoptosis and reactive oxygen species (ROS) production [23]. Another molecular docking and dynamics simulation assay predicted an antiviral role of hypericin in SARS-CoV-2 through targeting 3CLpro [8,9,24]. Herein, we aim to explore whether hypericin could serve as an antiviral compound for α-CoV.

## 2. Materials and Methods

### 2.1. Cell Lines, Virus, Reagents, and Antibodies

Vero cells [25] and swine testis (ST) cells [26] were maintained in Dulbecco՚s modified Eagle՚s medium (DMEM; Gibco, Grand Island, NY, USA) supplemented with 10% fetal bovine serum (FBS; Thermo Fisher Scientific, Waltham, MA, USA) and 1% penicillin-streptomycin (Solarbio, Beijing, China).

The PEDV HLJ strain (provided by Veterinary Pathology Laboratory, College of Veterinary Medicine of Northeast Agricultural University, Harbin, China) and TGEV Purdue strain [26] were propagated in Vero and ST cells, respectively.

Hypericin (Sigma, St. Louis, MO, USA) was dissolved with dimethylsulfoxide (DMSO; Solarbio) to prepare a stock of 200 mM and stored at −20 °C [23]. Ribavirin (Sichuan Baili Pharmaceutical Co., Ltd., Chengdu, China) was dissolved in DMEM to prepare a stock of concentration 200 mM that was used as a positive control for PEDV and TGEV inhibitor in our study [27].

Fluorescein isothiocyanate (FITC)-conjugated affiniPure goat anti-mouse IgG was purchased from the Zhongshan (Zhongshan, Beijing, China). The PEDV N protein monoclonal antibody [28] and the TGEV N protein polyclonal antibody were prepared and kept in Veterinary Pathology Laboratory, College of Veterinary Medicine of Northeast Agricultural University, China. The β-actin monoclonal antibody and horseradish peroxidase (HRP) labeled secondary antibodies were purchased from Sigma (Sigma, St. Louis, MO, USA).

### 2.2. Cytotoxicity Assay

Cells were seeded in 96-well plates at a density of 1.0 × 10^4^ cells/well, followed by additional serial dilutions (2.5–320.0 µM) of hypericin. Cells were allowed to grow for 48 h at 37 °C, and proliferation was analyzed using the Cell Counting Kit-8 (CCK-8; MedChemExpress, Monmouth Junction, NJ, USA) method. Briefly, the medium was removed before adding 10 µL of CCK-8 to the cells and incubating cells at 37 °C for 2 h. Absorbance of each well at OD_450_ was measured using an Elx 808 microplate reader (BioTek, Vermont, VT, USA), and the cell viability was calculated according to the following equation: Cell viability = (OD_a_ − OD_b_)/(OD_c_ − OD_b_) × 100%. OD_a_: Wells were incubated with hypericin treated cells and CCK8, OD_b_: Wells were incubated with DMEM (containing the same concentration of DMSO) and CCK8 but without cells, and OD_c_: wells were incubated with DMEM, containing the same concentration of DMSO-treated cells and CCK8.

### 2.3. Antiviral Assays on PEDV and TGEV

To evaluate the antiviral effect of hypericin in PEDV-infected Vero cells, we used two ways of treatment: pre-treatment and post-treatment (Figure 1A). For pre-treatment, Vero cells were initially treated with indicated concentrations of hypericin for 2 h at 37 °C and then inoculated with PEDV (0.1 MOI) for 36 h. For post-treatment, Vero cells were inoculated with PEDV (0.1 MOI) for 2 h at 37 °C and then added the indicated concentrations of hypericin for 34 h. Cells were finally harvested and prepared for further processing.

To assess the antiviral effect of hypericin in TGEV-infected ST cells, ST cells were inoculated with TGEV (0.1 MOI) for 2 h and then added the indicated concentrations of hypericin for 34 h. Cells were finally harvested and prepared for further processing.

### 2.4. Immunofluorescence Assay

Vero cells were fixed with 4% paraformaldehyde in phosphate-buffered saline (PBS) (Solarbio) followed by incubating with blocking buffer (0.2% skim milk, 2% FBS, 0.1 M glycine, 1% bovine albumin, and 0.01% triton-X 100 in PBS) for 30 min after incubating with PEDV N protein monoclonal antibody (1:500) for 1 h, followed by the incubation with fluorescence-conjugated goat anti-mouse IgG (1:1000) (Zhongshan, Beijing, China) and 2-(4-Amidinophenyl)-6-indolecarbamidine dihydrochloride (DAPI) (Solarbio) for 30 min. Fluorescence signals were captured with a Ti-S fluorescence microscope (Nikon, Tokyo, Japan). The percentage of viral infected cells was calculated by counting the cells with PEDV N-positive signal and dividing it by the number of DAPI-positive [10]. For each condition, we chose at least three views with more than 300 DAPI-positive cells and triplicated them to get the standard deviation.

### 2.5. Quantitative Real-Time PCR Assay

The total RNAs were extracted from the cells in a 6-well plate using Trizol reagents (Thermo Fisher Scientific) according to standard manufacturer’s protocol. The total RNAs were reverse transcribed into cDNAs using cDNA synthesis kit (Thermo Fisher Scientific) and stored at −20 °C. The mRNA expression levels of PEDV N gene and TGEV N gene were determined with an Applied Light Cycler 96 qRT-PCR System (Roche, Basel, Switzerland). The primers of PEDV N gene (F: GCACTTATTGGCAGGCTTTGT/R: ATTGAGAAAAGAAAGTGTCGTAG), β-actin (F: CTTAGTTGCGTTACACCCTTTC/R: TGTCACCTTCACCGTTCCA), TGEV N gene (F: TGACACTGACCTCGTTGCCAATG/R: CGTGACTTCTATCTGGTCGCCATC), and β-actin (F: CTTCTTGGGCATGGAGTCC/R: GGCGCGATGATCTTGATCTTC) were synthesized by Sangon Biotech Co., Ltd. (Shanghai, China). The expression levels of PEDV N gene and TGEV N gene were normalized to that of β-actin and then calculated using the 2^−ΔΔCt^ method [29].

### 2.6. Western Blot Assay

The proteins were harvested from cells in a 6-well plate through a protein extraction kit (Solarbio). Equal protein amounts were separated by sodium dodecyl sulfate polyacrylamide gel electrophoresis (SDS-PAGE), and after Western blot analysis, proteins were detected using specific antibodies against PEDV or TGEV N protein and β-actin and the Enhanced Chemiluminescence (ECL) Detection System (Clinx Science Instruments, Shanghai, China). PEDV or TGEV N protein signal intensities were normalized to β-actin and quantified using Image J software (National Institutes of Health, Bethesda, MD, USA) [6].

### 2.7. TCID50 Assay

We harvested the supernatants from 6-well plates and discarded the cell debris with 3000 g centrifugation for 5 min. The viral titers were determined with a diluted factor of 10, and the culture infectious dose (TCID_50_) units per mL of supernatant were calculated according to the Reed–Muench method [30]. TCID_50_/mL values were then converted into PFU/µL using the following equations: TCID_50_/mL × 0.00069.

### 2.8. Molecular Docking

The three-dimensional (3D) structure of hypericin was obtained by Chemdraw 19.0 program (Cambridge Software Company, Cambridge, MA), and the crystal structure of PEDV 3CLpro (PDB: 4XFQ) and TGEV 3CLpro (PDB: 2AMP) was from the Protein Data Bank (https://www.rcsb.org, accessed on 16 April 2021).

We performed a molecular docking analysis between hypericin and the dimer/monomer of PEDV 3CLpro/3CLpro^mut^ (Met6Ala, Ile151Ala, Asn152Ala, and Gln295Ala in dimer and Asn141Ala, Ile164Ala, Glu165Ala, Asp186Ala, Gln187Ala, and Gln191Ala in monomer) with AutoDock 4.2 (version 4.2.6) to predict the binding of hypericin to PEDV 3CLpro [31]. Briefly, we used 3CLpro/3CLpro^mut^ monomer–dimer structure as a receptor (corrected it with removing co-crystallized waters and adding charges and hydrogens) and hypericin as a ligand (adding charges). For dimer analysis, with a grid point spacing 0.500 Å, a grid box around selected active site residues was generated, where the center was at X: 25.274, Y: −0.089, and Z: −13.066 and the dimensions of the grid box were at X: 126, Y: 126, and Z: 126 (unit of the dimensions, Å). For monomer analysis, a grid point spacing was 0.400 Å, the center in the grid box was at X: 33.279, Y: 9.389, and Z: −12.570, and the dimensions were at X: 126, Y: 126 and Z: 126. The docking screening was performed by using the Lamarckian genetic algorithm and the default parameters of the genetic algorithm, except using a population size of 300 and a number of GA runs of 10. The estimated free energy of binding was determined and displayed in kcal/mol, and the final docking results were visualized and analyzed by using PyMOL 1.8 (Schrödinger, Inc., New York, NY, USA).

The molecular docking was also performed by AutoDock Vina (The Scripps Research Institute, San Diego, CA, USA) with a more specific grid box and center setting [32]. In dimer analysis, the grid point spacing was 0.375 Å, the center in the grid box were X: 35.718, Y: −4.454, and Z: −6.896, and the dimensions of the grid box were X: 52, Y: 54, and Z: 54. In monomer analysis, the grid point spacing was 0.499 Å, the center in the grid box was X: 25.539, Y: 25.108, and Z: −6.244, and the dimensions were X: 40, Y: 40, and Z: 40. The docking results were visualized and analyzed with PyMOL.

The molecular docking analysis between hypericin and TGEV 3CLpro (PDB: 2AMP) was performed with AutoDock 4.2 (version 4.2.6). In dimer analysis, the grid point spacing was 0.500 Å, a grid box around selected active site residues was generated, where the center was X: −31.798, Y: 18.004, and Z: −26.174, and the dimensions of the grid box were X: 126, Y: 126, and Z: 126. In monomer analysis, a grid point spacing was 0.500 Å, the center in the grid box was X: −20.841, Y: 9.571, and Z: −23.598, and the dimensions were at X: 100, Y: 100, and Z: 120.

### 2.9. Protein Expression and Purification

The gene of PEDV 3CLpro with a PreScission Protease site was cloned into pGEX-6p-1 vector [6] using *Bam*HI and *Xho*I restriction enzymes (TaKaRa Bio Inc., Shiga, Japan). The resulting pGEX-6p-1-3CLpro plasmid was transformed into competent Rosetta 2 (DE3) (Weidi Biotechnology, Shanghai, China). *Escherichia coli* (*E. coli*) strain and a single colony were used to inoculate 20 mL of Luria-Betani (LB) medium, supplemented with 50 µg/mL ampicillin before growth at 37 °C. After 16 h, the pre-culture was added to 2 L of LB medium (supplemented with 50 µg/mL ampicillin) and incubated at 37 °C in a shaking incubator (180 rpm). Expression of the fusion protein was induced by additional 1 mM isopropyl β-D-thiogalactoside (IPTG) (Solarbio) when the culture OD_600_ reached 0.7. At this point, the culture was transferred to a 16 °C shaking incubator (180 rpm) for an overnight culturing, and the bacteria were harvested by centrifuging at 5000× *g* for 10 min. Pellets were resuspended in lysis buffer (20 mg/mL lysozyme, 0.5 mM EDTA, 10% Triton-X, 20 mM Tris-HCl, pH 8.0) on ice, lysed by sonication in ultrasonic crusher (Scientz, Ningbo, China) for 30 min at 4 °C. The lysate was cleared with another centrifugation at 12,000× *g* for 20 min. The supernatant was packed into Glutathione Sepharose (GSH) 4B column (GE Healthcare, Chicago, IL), washed with 5 column volumes of washing buffer (PBS, pH 7.0), and subsequently eluted in the elution buffer (10 mM glutathione in 50 mM Tris-HCl, pH 8.0) (Solarbio). PreScission Protease (Beyotime, Shanghai, China) was added to the eluted fraction (1:10 *w*/*w*) to cleave the GST tag and finally packed the mixture into a GSH 4B column again to separate the 3CLpro from GST protein and protease, and harvest 10 mL 3CLpro with a concentration of 0.25 mg/mL.

### 2.10. PEDV 3CLpro Enzymatic Assay

The in vitro 3CLpro enzymatic assay to assess the hypericin inhibition was set up through fluorescence resonance energy transfer (FRET) (Olympus, Tokyo, Japan). The polypeptide substrate Dabcyl-YNSTLQ↓AGLRKM-E-Edans [33] was designed and synthesized by Sangon Biotech Co., Ltd. (Sangon) to measure PEDV 3CLpro activity. The cleavage of the FRET substrate was monitored by measuring the increase of fluorescence in a FLUOstar Omega plate reader (BMG labtech) at excitation and emission wavelengths of 340 nm and 485 nm, respectively. Total volume for the assay was 20 μL, containing 0.50 μg/mL, 1.00 μg/mL, or 2.00 μg/mL PEDV 3CLpro and 10 μM substrate in the assay buffer (20 mM Tris-HCl, pH 7.5). The reaction was monitored for 60 min immediately after adding the FRET substrate. To determine the inhibitory activity of hypericin, the enzyme was incubated with three different concentrations of hypericin (2.5 μM, 5.0 μM, and 10.0 μM) for 20 min at 37 °C. The system without hypericin but has an equal amount of DMSO (0.05% *v*/*v*) was used as a control, and the inhibition rate of hypericin on 3CLpro was calculated as follows: inhibition rate (%) = [1-experimental group fluorescence (60–0 min)/control group fluorescence (60–0 min)] × 100% [34,35].

### 2.11. Alignments of CoVs 3CLpro in Protein Sequence and Crystal Structure

The 3CLpro amino acid sequences of human coronavirus NL63 (HCoV-NL63) (Accession number: ABE97129.1), human coronavirus 229E (HCoV-229E) (Accession number: AGT21366.1), transmissible gastroenteritis virus (TGEV) (Accession number: P0C6Y5.1), SARS-CoV-2 (Accession number: YP_009725295.1), IBV (Accession number: AAZ82016.1), and PEDV (Accession number: MN644470.1) were achieved from National Center for Biotechnology Information (NCBI) (https://www.ncbi.nlm.nih.gov/, accessed on 29 April 2021), and HCoV-NL63, HCoV-229E, TGEV, and PEDV were further aligned with the Clustal Omega tool (https://www.ebi.ac.uk/Tools/msa/clustalo/, accessed on 29 April 2021). In addition, the crystal structure of PEDV 3CLpro (PDB: 4XFQ), TGEV 3CLpro (PDB: 2AMP), SARS-CoV-2 3CLpro (PDB: 6M2N), and IBV 3CLpro (PDB: 2Q6F) were obtained from the Protein Data Bank (https://www.rcsb.org, accessed on 16 April 2021) and further aligned with PyMOL.

### 2.12. Statistical Analysis

All the experiments were repeated three times, and all the data were expressed as the means ± standard deviation (SD). The statistical data were analyzed with SPSS 17.0 (SPSS Inc., Chicago, IL, USA) by using one-way analysis of variance (ANOVA) or Student’s *t*-test. *p*-values of <0.05 were considered as statistically significant, and *p*-values of <0.01 were considered as highly significant.

## 3. Results

### 3.1. Hypericin in PEDV-Infected Cells

Recently, we have shown that hypericin has an antiviral effect in IBV through apoptosis and ROS inhibition [23]. Simultaneously, another two groups utilized the molecular docking and dynamic stimulation that found hypericin precisely binds to SARS-CoV-2 3CLpro catalytic site [8,9], indicating an antiviral effect of hypericin in SARS-CoV-2 infection. Importantly, IBV and SARS-CoV-2 belong to the γ-CoV and β-CoV, respectively. These results encouraged us to assess whether hypericin has a broad antiviral effect in different CoV sub-families; we thus explored the effect of hypericin on PEDV, an α-CoV. With a CCK8 assay, we determined the 50% cytotoxic concentration (CC_50_) of hypericin in Vero cells was 56.73 ± 9.4 μM, and the maximal non-toxic concentration of hypericin was 10.0 μM, and we used this concentration as the highest dose for further cell studies (Figure 1B).

As an initial analysis, we processed two hypericin treatments, i.e., pre-treatment and post-treatment in PEDV-infected cells (Figure 1A). The pre-treatment of hypericin in PEDV-infected cells offers the information that whether hypericin has an effect on viral entry, which would further block viral infection, while the post-treatment indicates whether hypericin could block viral replication. Thus, hypericin (2.5 μM, 5.0 μM, and 10.0 μM) were added to the Vero cells with pre-treatment or post-treatment as described in Material and Methods before processing the cells for either quantitative real-time PCR (qRT-PCR) assay to assess viral N gene replication or TCID_50_ assay to evaluate the virion egression. As a positive control, ribavirin can block virus entry [36], and herein, we observed a block of PEDV entry into Vero cells in ribavirin pre-treatment condition with a significant decrease of PEDV-N gene and the viral titers (Figure 1C,D). Moreover, ribavirin is also an inhibitor of CoV protein and gene synthase [27], and it was therefore also used as a positive control in the post-treatment approach and, as expected, it showed a strong reduction of PEDV replication and virion egression (Figure 1C–H). In contrast to ribavirin, the pre-treatment of hypericin showed no inhibition of both the PEDV-N gene and the viral titers, whereas the post-treatment showed significant inhibition with both assays, especially the 10 μM of hypericin treatment to an extent was similar to ribavirin treatment (Figure 1C). The half-maximum effective concentration (EC_50_) of hypericin was calculated by the PEDV N gene expression (Figure 1C), which was 3.53 ± 0.33 μM.

In order to further confirm the inhibitory effect of hypericin on PEDV intracellular activities, we further examined the viral protein expression using other assays, i.e., viral N protein expression by Western blot and the number of viral N-positive cells by immunofluorescence (IF) in hypericin post-treated condition. Both assays showed that hypericin displayed a significant inhibition on viral N protein expression (Figure 1E–H). Moreover, a series concentration of hypericin ranging 0.625–10.0 μM confirmed the antiviral effect of hypericin was dose-dependent (Figure 1G,H). Taken together, our results showed that hypericin can efficiently inhibit PEDV infection.

### 3.2. Molecular Docking of Hypericin onto PEDV 3CLpro

As hypericin is precisely bound to the SARS-CoV-2 3CLpro catalytic site (Cys145) [8,9], we initially aligned the protein sequences and the 3D structures of 3CLpro of PEDV and SARS-CoV-2. Interestingly, we found they showed highly superimposed structures; in particular, the catalytic sites in SARS-CoV-2 3CLpro (His41 and Cys145) were the same as in PEDV (His41 and Cys144 in red) (Figure 2A,B). Although hypericin shows an antiviral effect in IBV through apoptosis and ROS inhibition [23], we also performed the same alignments between PEDV and IBV 3CLpro, and the catalytic sites were conserved (in green) (Appendix A). 3CLpro is a key viral protease that is essential for viral replication [7]. We also found hypericin could block viral replication through a significant inhibition of PEDV N gene expression (Figure 1C); this might suggest that hypericin has the high possibility to target the PEDV 3CLpro with the same molecular mode-of-action as in SARS-CoV-2.

We next moved to the detailed computational docking of PEDV 3CLpro and hypericin to explore whether and how hypericin targets the PEDV 3CLpro. As PEDV 3CLpro (PDB: 4XFQ) is resolved in dimeric form and the resolution is 1.65 Å [33], we further employed 3CLpro dimer or monomer as a receptor and hypericin as a ligand using AutoDock 4.2. Interestingly, we got two promising solutions with extremely high binding affinities of −9.4 kcal/mol (in dimer, solution 1) and −9.9 kcal/mol (in monomer, solution 2) from each of the 50 solutions (Figure 3A,B). Hypericin bounds to the 3CLpro dimer binding surface and interacts mainly with the B chain of 3CLpro; briefly, the main chains of Ile151 (distance: 3.0 Å) form hydrogen bonds with 8-OH in hypericin, and the side chains of Asn152, Gln295, and Met6 (distance: 2.4, 2.9, and 3.1 Å) interact with 8-OH, 13-OH, and 14-C=O, 1-OH in hypericin through hydrogen bonds (Figure 3C). Meanwhile, Met6 and Ile151 in the dimer B chain and Ala122 in the dimer A chain stabilized the conformation of hypericin by hydrophobic interaction (Figure 3D). Overall, this allowed hypericin to fit into a specific binding pocket on the 3CLpro dimer. Moreover, we also found hypericin bound to the 3CLpro monomer via another binding pocket in which the main chains of Asp186, Glu165, and Gln187 (distance: 2.5, 2.7, and 2.5Å) interacted with the 8-OH, 3-OH, and 4-OH, 6-OH in hypericin through hydrogen bonds. The side chains of Asn141 and Gln191 (distance: 2.9 and 3.5 Å) formed hydrogen bonds with the 1-OH and 6-OH of hypericin (Figure 3E), and the Leu164 further stabilized the conformation of hypericin through hydrophobic interaction (Figure 3F). In order to verify these computational docking in AutoDock 4.2, we conducted another molecular docking using AutoDock Vina. Consistently, we found that hypericin bound to 3CLpro dimer and monomer via the same pockets and residues (Appendix A).

To further validate the binding pockets in solutions 1 and 2, we performed another computational docking with the solution 1 mutant (Met6Ala, Ile151Ala, Asn152Ala, and Gln295Ala) and solution 2 mutant (Asn141Ala, Ile164Ala, Glu165Ala, Asp186Ala, Gln187Ala, and Gln191Ala). As shown in Figure 3G–J, neither of these two mutants displayed a promising pocket. Herein, our results suggested that hypericin has the high potential to bind to PEDV 3CLpro via at least two pockets.

### 3.3. In Vitro 3CLpro Enzymatic Inhibition by Hypericin

Although hypericin targeted the catalytic site (Cys145) of SARS-CoV-2 3CLpro [8,9], we did not observe a direct binding of hypericin to PEDV 3CLpro catalytic site (Cys144). However, we found this catalytic site (Cys144) was next to our solution 2 (Figure 3B). To further determine whether hypericin could inhibit the enzymatic activity of 3CLpro, we took advantage of an established FRET assay [37,38,39] by using the reporter FRET substrate Dabcyl-YNSTLQ↓AGLRKM-E-Edans. As a start, we expressed the recombinant PEDV 3CLpro with conventional methods of prokaryotic expression, followed by purification with Glutathione Sepharose (GSH) 4B column (GE Healthcare, Chicago, IL, USA) (Figure 4A). To confirm this system, we first added three different concentrations of 3CLpro (0.50 μg/mL, 1.00 μg/mL, and 2.00 μg/mL) into the substrate and found the fluorescence activity increased through the extension time, indicating that the purified 3CLpro could effectively cleave the fluorescent peptide substrate and chose 1.00 μg/mL of 3CLpro for further experiments (Figure 4B). Next, we added a series of concentrations of hypericin into the system and found that hypericin showed a high inhibition of PEDV 3CLpro cleavage activity with a half-maximal inhibitory concentration (IC50) of 5.90 ± 0.26 μM (Figure 4C).

These results, in agreement with the docking predictions, indicate that hypericin is a 3CLpro inhibitor.

### 3.4. The Hypericin-3CLpro Binding Domains Are Highly Conserved in α-CoV

To investigate whether the antiviral effect of hypericin is specific for PEDV or also in other α-CoVs, we aligned four α-CoV 3CLpro sequences, including PEDV, TGEV, HCoV-229E, and HCoV-NL63. We found these 3CLpro sequences were highly conserved and shared more than 60% identity with PEDV 3CLpro (Appendix A), especially the residues in solution 1 (Met6, Ala122, Ile151, Asn152, and Gln295) and solution 2 (Asn141, Leu164, Glu165, Asp186, Gln187, and Gln191) (Figure 3 and Figure 5A), suggesting that hypericin might also be able to inhibit other α-CoV 3CLpro.

TGEV is a porcine α-CoV that is often accompanied by mixed infection with PEDV, which seriously concerns the pig industry, and there is no effective therapy yet [40,41]. In order to explore whether hypericin could be a good therapy candidate for both viruses, we analyzed the available crystal structure of TGEV 3CLpro (PDB: 2AMP) and aligned it with PEDV. As shown in Figure 5B, these two 3D structures of 3CLpro were highly overlapped, especially the two pockets. To further confirm the molecular mode-of-action of hypericin to TGEV 3CLpro, we performed the docking again and found hypericin bound to TGEV 3CLpro dimer and monomer via the same pockets and similar residues as in Figure 3, i.e., Met6, Ala7, Ser110, Phe111, Gly122, Val150, Leu151, and Glu152 in the dimer (−10.15 kcal/mol) and Thr47, Thr48, Leu164, Glu165, Leu166, Gly167, Gln187, Met190, and Gln191 in the monomer (−10.17 kcal/mol) (Appendix A).

Overall, the amino acids located in solution 1 and solution 2 that mediated the binding of hypericin onto TGEV 3CLpro were highly conserved with similarity 63.6% as PEDV, indicating that hypericin might inhibit TGEV through similar motifs in 3CLpro.

### 3.5. Antiviral Activity of Hypericin on TGEV

To further confirm the potential effect of hypericin on TGEV, we decided to check the antiviral effect of hypericin in TGEV-infected ST cells. As have done in Figure 1, we found the CC_50_ of hypericin in ST cells was 97.06 ± 9.4 μM, and the maximum non-toxic concentration was 10.0 μM (Figure 6A). The same concentrations of hypericin (2.5, 5.0, and 10.0 μM) were thus used as in Figure 1.

ST cells were infected with TGEV at an MOI of 0.1 and post-treated with indicated concentrations of hypericin at 2 hpi, and then the cells were further processed for TCID50 to test the release of mature virions. Similar to PEDV, we found that hypericin could inhibit virion egression in a dose-dependent manner (Figure 6B). In order to confirm the antiviral effect of hypericin was through the inhibition of viral replication, we processed the samples to qRT-PCR and Western blot to check both the viral N gene and N protein expression. Consistent with PEDV, hypericin efficiently inhibits the TGEV N gene expression and protein expression (Figure 6C–E), and the EC_50_ of hypericin that was calculated by qRT-PCR assay was 2.11 ± 0.14 μM. This indicates that hypericin impaired TGEV infection through the inhibition of the viral intracellular replication, which might owe to the direct binding of hypericin onto TGEV 3CLpro.

## 4. Discussion

Hypericin has been shown to have a broad inhibitory effect on a variety of viruses [15,22,42,43], in particular IBV, a γ-CoV [23]. This prompted a number of groups to investigate whether hypericin has an effect on the current CoV disease 2019 (COVID-19) pandemic and eventually to repurpose this known compound [8,9,10]. We, however, focused on veterinary science and thus explored the antiviral effect of hypericin in porcine α-CoV, especially PEDV and TGEV, that caused huge losses in the global pig industry [44]. In this study, we assessed the antiviral activity of hypericin in PEDV- and TGEV-infected cells through viral mRNA/protein expression and virion and found that hypericin inhibits both PEDV and TGEV infection in cells (Figure 1 and Figure 6).

We have previously shown that hypericin inhibits γ-CoV through the reduction of viral-induced ROS and apoptosis, which in turn inhibited the viral replication [23]. However, several molecular docking assays showed that hypericin binds to the main intracellular replication enzyme (3CLpro) of SARS-CoV-2, indicating that hypericin has the potential to target β-CoV 3CLpro [8,9,45]. It remained to explore if hypericin could serve as a viral inhibitor for PEDV and TGEV as 3CLpro sequences and structures in CoV are highly conserved shared 51% averaged sequence identity (Figure 2, Figure 5 and Appendix A) [7], and they are involved in viral intracellular replication through the self-cleavage of viral pp1a and pp1ab. We thus evaluated the viral replication upon hypericin treatment in PEDV- and TGEV-infected cells via the expression levels of viral N gene and also monitored viral protein translation via N protein expression and virion egress through titration assay (Figure 1 and Figure 6). The viral inhibition was observed not only in translation and release but also in replication, which indicated that hypericin targeted the intracellular replication. In contrast to our previous γ-CoV study [23], we did not study the apoptosis and oxidative stress in PEDV or TGEV here due to the fact that the direct binding of hypericin onto 3CLpro was reported in SARS-CoV-2 [8,9,10], which guided us to validate whether hypericin target to TGEV and PEDV 3CLpro (Figure 3, Appendix A). As a consequence, the apoptosis and oxidative stress induction shown in the γ-CoV study might be a secondary effect of 3CLpro inhibition.

Computational docking is a method of drug design based on the characteristics of receptors and the interaction between receptors and drug molecules [46]. It is a theoretical simulation to study the interaction between ligand and receptor and predict their binding mode and affinity [47]. Moreover, the computational docking is accompanied by FRET assay, which is an effective mean of exploring whether the molecules could be inactive the enzymatic protein and is regarded as an important technology in the development of anti-CoV compounds [45]. Thus, our FRET results (Figure 4) confirmed the binding of hypericin onto PEDV 3CLpro (Figure 3 and Appendix A), leading to an inhibitory effect. Moreover, it has been reported that PEDV and TGEV 3CLpro form four specific druggable pockets (named S1, S1′, S2, and S4) [33,48]. Specifically, His41 and Cys144 were the key amino acids that have catalytic activity in PEDV and TGEV 3CLpro and form the S1 pocket together with Phe139, Ile140, Gly142, Ala143, His162, Gln163, Glu165, and His171 [33,48]. Meanwhile, several studies reported that natural compounds in traditional herbs could bind to the 3CLpro of CoV, leading to the inhibition of viral replication [49,50,51,52,53]; in particular, the natural compound, quercetin, inhibited the PEDV 3CLpro through Cys144, Asn141, and His162 that were located in S1 [33,54]. In this study, we found the Asn141 and Glu165 in solution 2 (Figure 3 and Appendix A) were part of this S1 pocket, indicating hypericin indirectly inhibit the catalytic activity of 3CLpro in PEDV. Additionally, targeting the catalytic site of viral protease might promiscuously inhibit the activity of other cellular proteases, which in turn could cause undesirable side effects. Since we found two binding pockets in PEDV 3CLpro and none of the target domains was the key catalytic residue, this minimized the possibility of interfering with other cellular proteases, which still need to be assessed in future work. In addition, in this study, according to the method of finding the optimal solution in the literature [32], we determined the optimal solution mainly from the estimated free energy of binding and the frequency of occurrence of hypericin similar conformations (Figure 3 and Appendix A).

Could hypericin be a pan-α-CoV inhibitor? The motifs that mediated the binding of hypericin onto PEDV 3CLpro were highly conserved in α-CoVs, especially in TGEV (Figure 5). The docking results further showed that hypericin might use the same druggable pocket to bind TGEV 3CLpro dimer and monomer as PEDV 3CLpro (Appendix A). Interestingly, hypericin bound to TGEV 3CLpro with a much lower binding energy (−10.15 kcal/mol in dimer and −10.17 kcal/mol in monomer), and the residues Thr47, Leu164, Glu165, Leu166, and Gln191, which mediated the binding of hypericin onto TGEV 3CLpro, were located at not only the reported S1 pocket but also at S2 (Pro188, Leu164, Ile51, Thr47, His41, and Tyr53) and S4 (Leu164, Leu166, Tyr184, and Gln191) pockets of TGEV 3CLpro [33,48,55]. Moreover, the EC50 of hypericin in TGEV-infected cells showed a significantly lower value (2.11 ± 0.14 μM) compared to the PEDV (3.53 ± 0.33 μM), indicating that hypericin exhibit more significant effect on TGEV 3CLpro than PEDV.

Although our study has demonstrated that hypericin could bind to α-CoV 3CLpro and inhibit its activity, the actual binding sites are further needed to firmly determine through co-crystallization of 3CLpro and hypericin. Obtaining this structural information can inform on possible chemical modifications to increase their binding affinity, allowing to lower the working concentration in vivo. Instead, our study determined hypericin serves as a pan-anti-α-CoV inhibitor in terms of targeting 3CLpro, as 3CLpro is the main protease to process the nsps for viral replication. Future research should focus on whether the inactivation of 3CLpro by hypericin affects the formation of CoV replication structures, e.g., double-membrane vesicles (DMVs) or convoluted membranes that lead to apoptosis or ROS inhibition, and further impairs the viral replication. This will pave the way to provide a theoretical mechanism to understand the process of CoV replication structural formation.

## 5. Conclusions

Hypericin is a novel anti-CoV compound that effectively inhibits the replication of PEDV and TGEV through 3CLpro. Our findings offer novel and promising therapeutic possibilities against infections caused by α-CoV, and hypericin deserves further attention as not only a pan-anti-α-CoV compound but potentially also as a compound of other coronaviral infections.

## Figures and Tables

**Figure 1 viruses-13-01825-f001:**
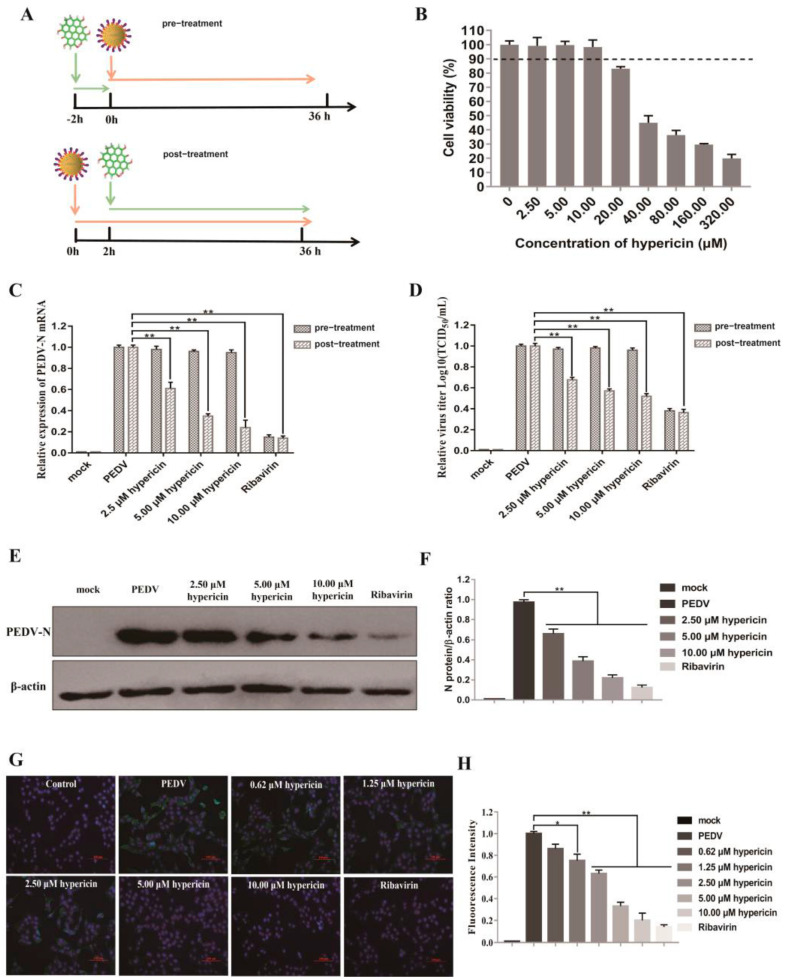
Hypericin in PEDV-infected cells. (**A**) An overview of hypericin pre-treatment and post-treatment in PEDV-infected Vero cells. (**B**) Cell viability upon hypericin treatment in Vero cells. Cells were treated with the indicated doses of hypericin for 48 h, and the cell viability was further measured by CCK-8 assay. All CCK-8 values were normalized to the 0 µg/mL, which represents 100% cell viability. (**C**) Vero cells were infected with PEDV at MOI 0.1 with hypericin pre-treatment and/or post-treatment at the concentrations of 2.5 µM, 5.0 µM, and 10.0 µM. The ribavirin treatment (200 μM) was a positive control to block PEDV entry or replication. The cells were processed for RNA extraction; viral N gene was quantified by qRT-PCR, using the primer sets described in Materials and Methods. The expression levels of viral N gene were normalized to those of β-actin, and data represent average amounts relative to the PEDV-infected cells with 0.05% (*v*/*v*) of DMSO treatment. (**D**) Culture supernatants were examined by TCID_50_. Results were expressed relative to the PEDV-infected cells with 0.05% (*v*/*v*) of DMSO treatment. (**E**) Proteins were separated by SDS-PAGE, and Western blot membranes were probed with an antibody against either PEDV N or GAPDH. (**F**) The N protein expression in each sample was quantified and normalized to the GAPDH signal. Results were expressed relative to the PEDV-infected cells with 0.05% (*v*/*v*) of DMSO treatment. (**G**) The post-treatment of hypericin in PEDV-infected cells at the concentrations of 0.625 µM, 1.25 µM, 2.5 µM, 5.0 µM, and 10.0 µM. Cells were processed for immunofluorescence using antibodies against PEDV N protein and DAPI staining. (**H**) The percentage of viral infected cells was calculated by counting the cells with PEDV N-positive signal and dividing it by the number of DAPI-positive. For each condition, we chose at least three views with more than 300 DAPI-positive cells and triplicated them to get the standard deviation. All data are represented as mean ± SD of at least three independent experiments. Student’s *t*-test was used to evaluate statistical differences, and a *p* < 0.05 was considered significant with *, and ** showed *p* < 0.01.

**Figure 2 viruses-13-01825-f002:**
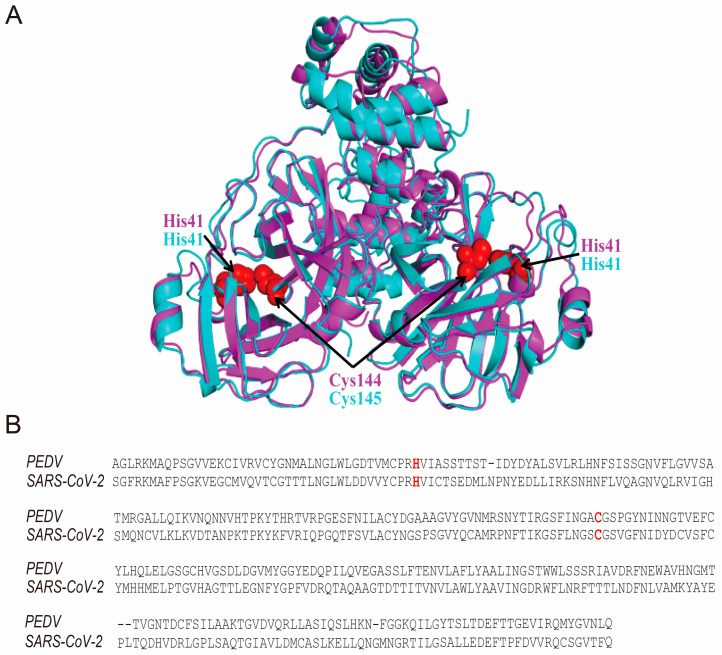
Alignments of PEDV and SARS-CoV-2 3CLpro in protein sequence and crystal structure. (**A**) The sequences of the PEDV 3CLpro (Accession number: MN644470.1) and SARS-CoV-2 3CLpro (Accession number: YP_009725295.1) are aligned by Clustal Omega tool (https://www.ebi.ac.uk/Tools/msa/clustalo, accessed on 29 April 2021). The catalytic sites in SARS-CoV-2 3CLpro (His41 and Cys145) and PEDV (His41 and Cys144) are shown as red. (**B**) The crystal structures of PEDV 3CLpro (PDB: 4XFQ) and SARS-CoV-2 3CLpro (PDB: 6M2N) are aligned by PyMOL. PEDV 3CLpro is shown as pink cartoon, and SARS-CoV-2 3CLpro is shown as blue cartoon. The catalytic sites in SARS-CoV-2 3CLpro (His41 and Cys145) and PEDV (His41 and Cys144) are shown as red spheres.

**Figure 3 viruses-13-01825-f003:**
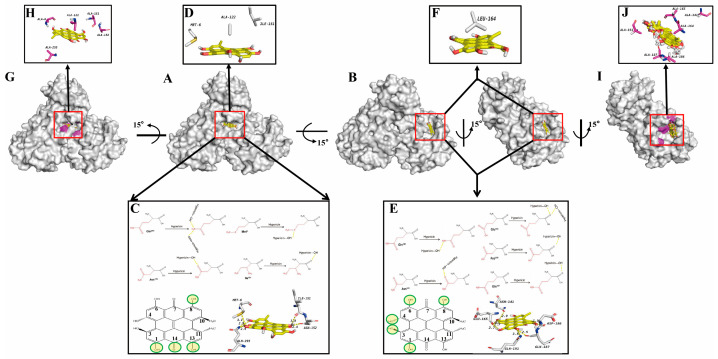
Molecular docking results of hypericin onto PEDV 3CLpro. The docking results were visualized by PyMOL. The main chains of amino acids are shown in black, and the side chains are shown in red. Hypericin is shown as yellow ligand, hydrogen bond is indicated as yellow dashed lines, and the atoms in hypericin interacting with amino acids are shown as yellow circles. The overview of solution 1 showing hypericin docked onto 3CLpro dimer with binding affinities of −9.4 kcal/mol (**A**) or solution 2 in 3CLpro dimer (left) and monomer (right) with binding affinities of −9.9 kcal/mol (**B**). The hydrogen bond interactions of hypericin with Ile151, Asn152, Gln295, and Met6 in 3CLpro dimer (**C**), and Asn141, Glu165, Asp186, Gln187, and Gln191 in 3CLpro monomer (**E**). (**D**) Hydrophobic interaction of Met6 and Ile151 in the dimer B chain and Ala122 in the dimer A chain with hypericin. (**F**) Hydrophobic interaction of Leu164 with hypericin in the monomer. (**G**,**H**) The docking results of hypericin with dimer of PEDV 3CLpro^mut^. (**G**) The overview of the docking of hypericin and dimer of PEDV 3CLpro^mut^. (**H**) Met6Ala, Ile151Ala, Asn152Ala, and Gln295Ala in dimer. (**I**,**J**) The docking results of hypericin with monomer of PEDV 3CLpro_mut_. (**I**) The overview of the docking of hypericin and dimer of PEDV 3CLpro^mut^. (**J**) Asn141Ala, Ile164Ala, Glu165Ala, Asp186Ala, Gln187Ala, and Gln191Ala. Hypericin is shown as yellow ligand, and the mutated residues are shown in pink.

**Figure 4 viruses-13-01825-f004:**
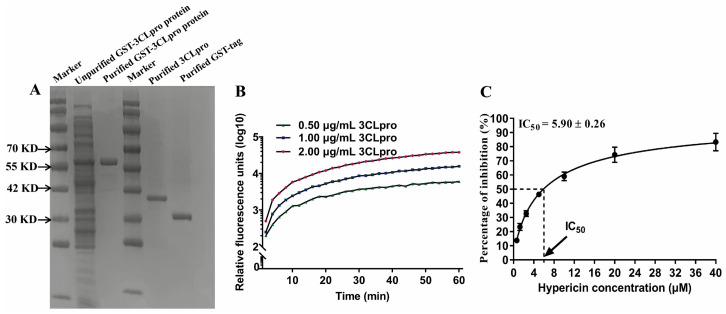
Hypericin impairs PEDV 3CLpro activity. (**A**) The expression and purification of PEDV 3CLpro. (**B**) The cleavage activity on FRET substrate Dabcyl-YNSTLQ↓AGLRKM-E-Edans by 0.50 μg/mL, 1.00 μg/mL, and 2.00 μg/mL of purified PEDV 3CLpro was monitored through the fluorescence value. (**C**) Inhibition of cleavage activity of 1.00 μg/mL PEDV 3CLpro in the presence of increasing concentrations of hypericin. All data are represented as mean ± SD of at least three independent experiments.

**Figure 5 viruses-13-01825-f005:**
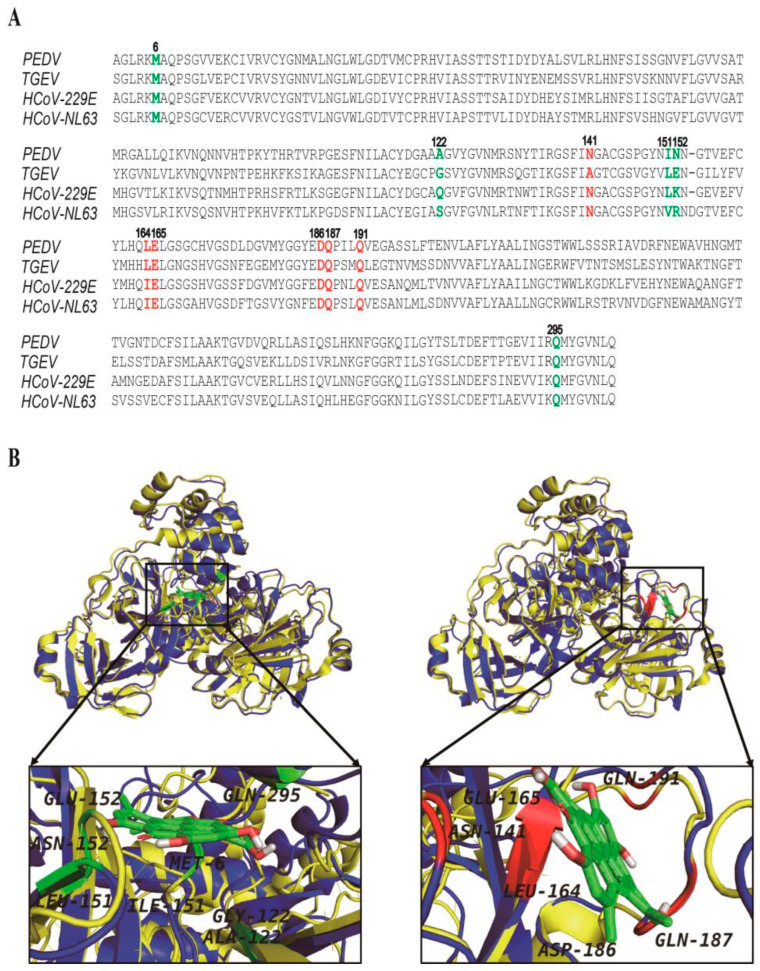
Hypericin targets the TGEV 3CLpro through the same region as PEDV. (**A**) The 3CLpro sequences of the PEDV (Accession number: MN644470.1), HCoV-NL63 (Accession number: ABE97129.1), HCoV-229E (Accession number: AGT21366.1), and TGEV (Accession number: P0C6Y5.1) are aligned by Clustal Omega tool (https://www.ebi.ac.uk/Tools/msa/clustalo, accessed on 29 April 2021). Solution 1 of hypericin targeting PEDV 3CLpro is shown in green, and solution 2 is in red. (**B**) The alignments of PEDV and TGEV 3CLpro 3D structures. The blue cartoon represents PEDV, and the yellow cartoon represents TGEV. Hypericin is shown as green ligand, and the residues mediating the binding of hypericin onto PEDV 3CLpro in solution 1 (**left**) and solution 2 (**right**) are shown.

**Figure 6 viruses-13-01825-f006:**
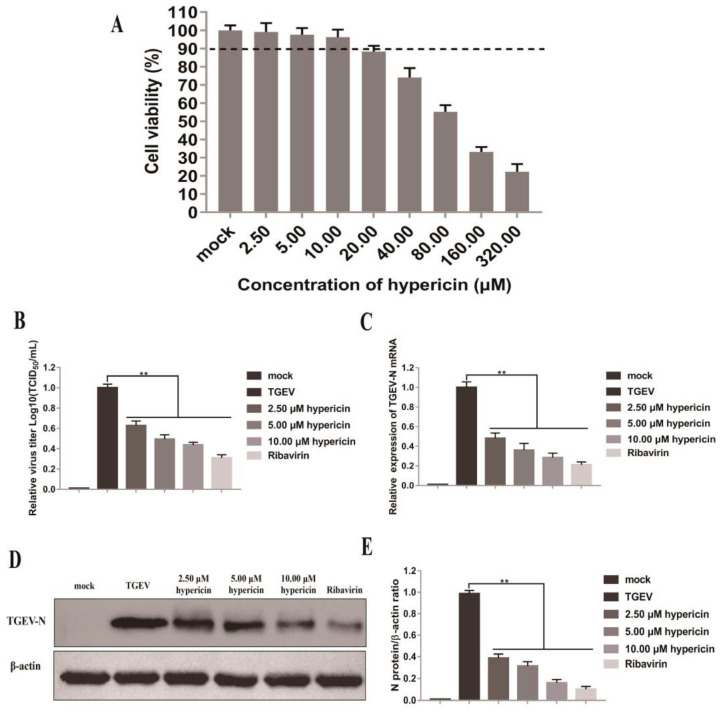
Hypericin in TGEV-infected cells. (**A**) The cell viability of hypericin in ST cells was tested by CCK-8 assay as Figure 1B. (**B**) ST cells were infected with TGEV at an MOI of 0.1 for 2 h and then treated with 2.5 μM, 5.0 μM, and 10.0 μM of hypericin for another 34 h. The ribavirin treatment was used as a positive control to block PEDV entry or replication. The culture supernatants were collected and processed for TCID50 assay and quantified as Figure 1D. (**C**) The cells were processed for RNA extraction, and viral N gene was quantified by qRT-PCR, using the primer sets described in Materials and Methods and quantified as Figure 1C. (**D**) Proteins were separated by SDS-PAGE, and Western blot membranes were probed with an antibody against either TGEV N or GAPDH. (**E**) The N protein expression in each sample was quantified and normalized as Figure 1F. All data are represented as mean ± SD of at least three independent experiments. Student’s *t*-test was used to evaluate statistical differences, and a *p* < 0.05 was considered significant with ** showed *p* < 0.01.

## Data Availability

The data generated and analyzed in this study are included in the article.

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
