# Peer review of "Hypericin Inhibit Alpha-Coronavirus Replication by Targeting 3CL Protease"

_viruses, 2021, doi:10.3390/v13091825_

Round 1

Reviewer 1 Report

The article written by Yue Zhang et al. describes hypericin as inhibitor of alpha-coronavirus replication by targeting 3CL protease. The authors used experimental and theoretical techniques. The work deals with the current problems related to the search for new antiviral drugs. The research is very interesting, I have no additional comments. Therefore, the work may be published in its current form.

Author Response

Dear reviewer,

Thank you for your unanimously and overwhelmingly positive on our manuscript, we hope our story brings more insight to develop the hypericin as a pan-anti-CoV compound.

Best regards, Yingying Cong

Reviewer 2 Report

In this manuscript, the authors are examining hypercine's ability to bind viral 3CLpro. Hypercin is already known to have anti-viral activity and is used in other treatments of viral infections. The authors are hypothesizing that hypercin will serve as an anti-viral agent for Alphacoronavirus (α-CoV) and other viral targets through assays and molecular docking calculations. 

The authors have aleady shown hypercin's antiviral effects (ref 23) as well as another group (ref 8 and 9). 

This reviewer does not believe that ths work is suitable to be published in Viruses. Perhaps a biologically-orientated computational journal would be a better fit. 

Furthermore, A crystal structure of the bound drug would be more noteworthy.

New insights are not clearly described in the abstract or conclusion leaving the signicance in question.

Author Response

Thanks to this reviewer's critical comments on our manuscript, we have generated a point-to-point response to the comments.

  1. This reviewer is correct that we have already shown the antiviral effect of hypericin in IBV (ref 23), as well as other groups in SARS-CoV-2 (ref 8 and 9), however, hypericin block the ROS and apoptosis in IBV, a γ-CoV (ref 23) and hypericin target the SARS-CoV-2 (a β-CoV) 3CLpro catalytic sites (ref 8 and 9). These data drive us to explore whether hypericin has an antiviral effect in α-CoV, in particular the PEDV and TGEV, which have caused huge losses in the global pig industry. Therefore, it is necessary to explore whether hypericin has inhibition on α-CoV through 3CLpro. In addition, our data could further confirm whether hypericin could be developed as a pan-anti-CoV therapy.
  2. Although this work involves some computer molecular dockings, they ultimately serve as a means of detecting to investigate hypericin’s antiviral effects on α-CoV, which completely fits the Viruses journal's field of antiviral drugs, antiviral therapy, and antiviral drug development, and thus we believe our work should be suitable to publish in Viruses.
  3. We agree with this reviewer that the crystal structure of hypericin bound to PEDV/TGEV 3CLpro would be more noteworthy and this is also what we are processing now as a follow-up study. We have mentioned the importance of the co-crystallization of 3CLpro and hypericin, this would inform us to further make the possible modification of hypericin to increase the binding affinity (lines 556-559, page 14).
  4. According to this reviewer’s suggestion, we add new insight in the abstract or conclusion (lines 31-33, page 1, and lines 567-572, page 14) that hypericin has the potential to inhibit α-CoV by targeting 3CLpro, especially with all the inconveniences that SARS-CoV-2 are currently bringing us, our project is again of great significance not only for α-CoV but also potentially pan-anti-CoV.

Reviewer 3 Report

The manuscript reports biological evaluation and molecular docking study of hypericin as anti-α-CoV agent. I would recommend that the authors reconsider the following points before publication in viruses.

The authors might want to add the methodology in the text how to specify the active sites of proteins, 4XFQ and 2AMP. Because AutoDock can’t predict the sites.

The expression “highly conserved” on pages 1, 2, 8, 10, 11, and 13 is unclear. It’s suitable for scientific papers to show the exact numbers of matching percentage of amino acids in total amino acid sequence that the authors discuss.

The conclusion section should be included in the main text.

Author Response

Thanks to this reviewer's critical comments on our manuscript, here is the reply for each comment.

  1. This reviewer is correct that AutoDock cannot predict the catalytic site of 3CLpro. However, as we mentioned in the manuscript (line 529-533, page 13) that the catalytic sites of PEDV and TGEV 3CLpro were already determined by other two studies (see ref 33 and 48), as a result, we did not perform any experiments to show the catalytic sites of PEDV 3CLpro (4XFQ) or TGEV 3CLpro (2AMP).
  2. We agree with the reviewer that the percentage of sequence similarity is necessary to address the conservation.

    It has been shown the average similarity of 3CLpro among CoVs was 51%, so we added it in the manuscript (line 60, page 2).

    As shown in Fig. 2B that the catalytic sites of 3CLpro in PEDV and SARS-CoV-2 are the same, so we corrected the “highly conserved” in lines 327-328, page 8 to “the same as”.

    We have added the protein identity between TGEV/HCoV-229E/HCoV-NL63 and PEDV 3CLpro in supplementary table 1 to show the conservation between α-CoV 3CLpro full length. Our modifications were in line 245, page 5, and 425-426, page 10, and 442, page 11.

    We corrected the “highly conserved” in the line of 504, page 13 to “shared 51% averaged sequence identity”, and we also added the ref7.

  3. We have added the conclusion section in lines 568-573, page 14.